# The Many Faces of DFNB9: Relating *OTOF* Variants to Hearing Impairment

**DOI:** 10.3390/genes11121411

**Published:** 2020-11-26

**Authors:** Barbara Vona, Aboulfazl Rad, Ellen Reisinger

**Affiliations:** Tübingen Hearing Research Centre, Department of Otolaryngology, Head & Neck Surgery, University of Tübingen Medical Center, 72076 Tübingen, Germany; barbara.vona@uni-tuebingen.de (B.V.); Aboulfazl.rad@uni-tuebingen.de (A.R.)

**Keywords:** DFNB9, otoferlin, sensorineural hearing loss, auditory synaptopathy/neuropathy, temperature-sensitive auditory neuropathy, progressive hearing loss

## Abstract

The *OTOF* gene encodes otoferlin, a critical protein at the synapse of auditory sensory cells, the inner hair cells (IHCs). In the absence of otoferlin, signal transmission of IHCs fails due to impaired release of synaptic vesicles at the IHC synapse. Biallelic pathogenic and likely pathogenic variants in *OTOF* predominantly cause autosomal recessive profound prelingual deafness, DFNB9. Due to the isolated defect of synaptic transmission and initially preserved otoacoustic emissions (OAEs), the clinical characteristics have been termed “auditory synaptopathy”. We review the broad phenotypic spectrum reported in patients with variants in *OTOF* that includes milder hearing loss, as well as progressive and temperature-sensitive hearing loss. We highlight several challenges that must be addressed for rapid clinical and genetic diagnosis. Importantly, we call for changes in newborn hearing screening protocols, since OAE tests fail to diagnose deafness in this case. Continued research appears to be needed to complete otoferlin isoform expression characterization to enhance genetic diagnostics. This timely review is meant to sensitize the field to clinical characteristics of DFNB9 and current limitations in preparation for clinical trials for *OTOF* gene therapies that are projected to start in 2021.

## 1. Introduction

Sensorineural hearing loss is one of the most common sensory deficits in humans, affecting one to two per 1000 newborns in developed countries [1]. Over the past 25 years since the discovery of the first deafness gene, more than 120 genes have been causally associated with non-syndromic hearing loss (https://hereditaryhearingloss.org/) and over 6000 disease-causing variants have been identified [2]. As most variants implicated in hearing loss are small insertions/deletions (indels) or single nucleotide variants [2], high-throughput sequencing is a well-suited method to rapidly allow for a deeper understanding of the spectrum of variants involved in deafness and their consequences on the auditory phenotype.

Using a candidate gene approach, the DFNB9 locus (OMIM: 601071) was mapped to chromosome 2p23.1 in 1996 by studying a genetically isolated family from Lebanon [3]. Three years later, the gene *OTOF* (OMIM: 603681), encoding a transmembrane (TM) protein called otoferlin, was mapped to the DFNB9 locus and identified as causing prelingual autosomal recessive, non-syndromic deafness [4]. Biallelic pathogenic variants in *OTOF* cause auditory synaptopathy due to deficient pre-synaptic neurotransmitter release at the ribbon synapse of the inner hair cells (IHCs) [5].

Since its initial identification, about 220 pathogenic and likely pathogenic variants in *OTOF* have been identified. In addition to an expanded understanding of the types of variants in otoferlin that cause deafness, the structure and function of otoferlin have been extensively characterized through functional studies that have greatly informed experimental therapies. This review covers the challenges of clinically diagnosing *OTOF*-associated hearing impairment, the spectrum of phenotypes that have been observed in patients with *OTOF* variants and a current review of genotype-phenotype correlations.

### 1.1. Mouse Studies Reveal Insights into Otoferlin Function

Otoferlin is distributed throughout the cytoplasm and plasma membrane of IHCs with the exception of the most apical part that forms the cuticular plate and tight junctions with neighboring cells (Figure 1). In addition, type I vestibular hair cells and immature outer hair cells (OHCs) express otoferlin, yet the physiological function for this expression in the mature inner ear is still unclear [6,7]. Although the mRNA of otoferlin can be isolated from several tissues including the brain, clear immunohistochemical proof of otoferlin protein expression outside hair cells is missing. Studies in *Otof*-knock-out mouse models revealed that, in the absence of otoferlin from IHCs, very few neurotransmitter-filled synaptic vesicles fuse with the plasma membrane [5,8]. Thus, acoustic stimuli still generate receptor potentials in the IHCs (and OHCs), but this information is not passed to the auditory pathway. In vitro studies indicating that otoferlin can interact with neuronal SNARE proteins contributed to the hypothesis that otoferlin acts as a synaptotagmin-like Ca^2+^ sensor for exocytosis [5,9]. However, later studies revealed that such neuronal SNAREs are expressed at only very low levels in IHCs and are absent from IHC synapses [10]. Instead, the mechanism of vesicle fusion might rely on a unique molecular mechanism in IHCs [11]. Later studies in a mouse line with the mutation of a presumed Ca^2+^-binding site revealed a slight delay and slowing down of Ca^2+^-triggered exocytosis, which would be in line with a Ca^2+^-dependent acceleration of exocytosis and was interpreted as a Ca^2+^ sensor function for otoferlin in exocytosis and vesicle replenishment [12]. However, the Ca^2+^-binding capability of the site targeted in this study is still under debate (see Section 5).

Notably, mouse models with reduced levels of otoferlin revealed additional functions for the protein at the synapse: In these models, Ca^2+^-triggered vesicle fusion still occurred, which allowed for the observation of synaptic processes that are closely linked to exocytosis. In this way, it was uncovered that the rate with which synaptic vesicles are regenerated, supplied to the active zones of the synapses, and rendered competent for Ca^2+^-triggered fusion depends on the quantity of otoferlin in the basolateral plasma membrane of IHCs [8,13]. A reduction to ~3% of otoferlin protein levels in the plasma membrane in the *pachanga* mouse model still enabled IHCs to release transmitter in response to short (<10 ms) stimuli, but strongly impaired the synaptic transmission for longer stimuli [8,13]. This was attributed to a defect in accomplishing vesicles competent for fusion—also termed “priming” or “vesicle replenishment to the readily releasable pool of vesicles”. As a result, in a living organism, such IHC synapses are constantly deficient of fusion-competent synaptic vesicles. Consequently, no auditory brainstem responses (ABRs) can be recorded in these animals [8,14], as this requires the synchronous action potential firing in the auditory pathway for hundreds of stimulus repetitions.

A milder reduction to 35% of wild-type otoferlin levels in the plasma membrane of a mouse model for the human p.Ile515Thr substitution halved the rate of vesicle replenishment compared to normal hearing controls [13]. This only mildly affected the auditory threshold but impaired the ability to detect changes on top of sustained stimuli. In this mouse line, stimulation of exocytosis resulted in enlarged synaptic vesicles. Together with the finding of otoferlin immunoreactivity on endosomal vesicular structures, it was concluded that otoferlin is involved in the reformation of synaptic vesicles from bulk endosomes [13]. This is in agreement with the finding that otoferlin interacts with the clathrin adaptor protein AP2 [15,16]. Presumably, otoferlin is retrieved from the plasma membrane mostly by bulk endocytosis. On the large endosomal structures, clathrin-coated pits appear, supposedly forming novel synaptic vesicles [13,16]. In conclusion, the proper function of the synapse being able to faithfully transmit highly fluctuating acoustic stimuli to the auditory pathway requires a high expression and proper localization of otoferlin. In contrast, low otoferlin levels allow for the transmission of acoustic signals as long as individual synapses are only sparsely activated.

### 1.2. Otoferlin Isoforms

The structural diversity of *OTOF* has been expanded since its identification to include long and short isoforms that make use of distinctive transcription and translational start sites, as well as alternative splicing of exons 6, 31, and 47. The short and long isoforms range from 28 exons spanning 21 kb [4] to 48 exons across 90 kb [17], respectively. In total, two long and three short isoforms have been identified in humans. Long isoforms are characterized by the presence of six (or seven) C_2_ domains and a C-terminal transmembrane (TM) domain, whereas the short isoforms are comprised of only the final three C_2_ domains and the TM domain [17]. C_2_ domains are globular domains composed of antiparallel β-sheets, which are known for Ca^2+^ and phospholipid binding. In humans, the short isoforms are comprised of isoform b (NP_004793) and d (NP_919304), each with 1230 amino acids and isoform c (NP_919303), with 1307 amino acids, employing an alternate starting exon, compared to the long isoforms a (NP_919224) and e (NP_001274418), which both encode 1997 amino acids.

With respect to a potential functional role of the short isoforms, a review of pathogenic and likely pathogenic variants has shown no indication that variants only affecting the long, but not short, isoforms would cause a milder phenotype. This confirms that the long isoform is critically required for normal hearing function [17].

The two long isoforms of otoferlin can be distinguished by virtue of tissue mRNA expression and subtle differences in exon usage at the 3′ end of the gene. Isoform a was identified from brain cDNA libraries with a termination codon in exon 47 [17]. An alternative splice isoform has been identified in the human cochlea that exclusively uses exon 48 to encode the C-terminus (isoform e, [18]), but lacks exon 47, a finding that was consistent with the mouse [17]. Moreover, pathogenic variants in exon 48, but not exon 47, indicate that the isoform that skips exon 47 and makes use of the termination codon in exon 48 seems to be the predominant isoform in the human cochlea [18,19,20].

Limitations in obtaining mRNA from human IHCs has presented a major bottleneck in profiling and quantifying the relative fractions of all otoferlin isoforms. According to Yasunaga et al. [17], an alternative splice acceptor site in exon 31 may be employed, eliminating 20 amino acids from the longest variant. This alternative splicing was predicted for the short isoform b and seems to be the predominant variant in mouse inner ear tissue [13]. In the presence of this 20 amino acid “RXR” motif, the p.Ile515Thr substitution caused a retraction of mouse otoferlin from the plasma membrane, which was not the case for the p.Ile515Thr-protein lacking the RXR motif. The authors of this study speculated that the phenotype found in patients with the p.Ile515Thr substitution would be best explained if the human cochlea expresses a mixture of both splice variants [13]. Furthermore, transcript analyses suggest the presence of so far undetected exons (described in Section 3.4 below). Despite these uncertainties, we recommend that human *OTOF* sequence analysis utilizes the reference sequence for variant e, NM_001287489, encoding the 1997 amino acid protein, NP_001274418. Furthermore, we propose that *OTOF* variants from human molecular genetic diagnostic laboratories that are deposited in clinical variant repositories be adjusted to this reference sequence.

## 2. Hallmarks of Audiometric Testing in DFNB9 Patients

Since its discovery, *OTOF*-associated hearing loss in humans has presented several challenges, making the selection of clinical diagnostic protocols an essential undertaking for an early diagnosis. Based on the finding that *OTOF* variants disrupt presynaptic function in IHCs rather than neuronal function, a change in terminology from “auditory neuropathy” to “auditory synaptopathy” has been adopted to more precisely describe this. A lesion to the neural pathway that transmits signals from the cochlea to the brain is clinically characterized by an absence of ABRs and the presence of otoacoustic emissions (OAEs). The testing of OAEs indicates proper functioning of cochlear amplification by the OHCs. As OHCs and IHCs employ the same protein machinery for mechanotransduction at their stereocilia bundle, any malfunction of this can be excluded if OAEs are present. Similarly, both hair cell types depend on proper endolymph composition and endocochlear potential as the driving force; a functional deficiency which would be detectable in altered OAE recordings. However, patients with *OTOF* variants lose OHC function, occasionally within the first year, and in about one-third of cases in the second year of life, with only few individuals displaying OAEs in early adulthood [20,21,22]. Therefore, individuals without OAEs should be considered for genetic testing that includes *OTOF*.

As is true for all forms of auditory neuropathy or synaptopathy, newborn hearing screening that tests for OAEs, e.g., with distortion product otoacoustic emissions (DPOAEs) or transient evoked otoacoustic emissions (TEOAEs), fail to detect a hearing disorder in most cases, as OAEs are initially present. Passing OAE tests despite profound deafness can be misleading and, in the worst-case scenario, prevent parents and pediatricians from pursuing a more complete audiological diagnostic testing. ABRs in patients with synaptopathies are typically absent, even for high sound pressure levels, making them well-suited for the detection of profound deafness. Moreover, even mild forms of hearing loss result in abnormal ABRs in case of DFNB9 (see below). However, as ABR testing is a more time-consuming procedure, this is not routinely applied in newborn hearing screening protocols, delaying the diagnosis of a baby with congenital auditory synaptopathy by months or even years until it is recognized and confirmed. In children with *OTOF* variants, behavioral audiometry, with or without visual reinforcement, can indicate severe to profound hearing loss across all frequencies. While some patients display residual hearing in the low-frequency region (with thresholds of ~75 dB hearing level (HL) for 250 Hz; e.g., [23]), pure tone audiograms may be flat, or bowl-shaped, but in all such cases, average thresholds are above 90 dB HL.

A reliable diagnosis of even mild forms of hearing loss caused by *OTOF* variants is of relevance, especially for young children whose speech acquisition may become strongly impaired. Moreover, a precise diagnosis will also help later in life to specify impairments of speech comprehension, which, for example, may explain why following multiple speakers is much more exhausting for DFNB9 listeners than for normal hearing listeners. Despite pure tone audiograms being only mildly or moderately affected in such cases, ABRs are mostly abnormal, indicating higher thresholds than expected from psychophysical testing. ABR waves I to III are hardly detectable and waves IV and V are delayed [24]. Speech comprehension testing should be performed both in silence and in background noise, the latter of which is typically strongly affected.

A more specific test for this type of synaptopathy would be to quantify the time required for synaptic regeneration. This could be done by gap detection tests, i.e., silent gaps of different length in broadband noise. Intact IHC synapses accurately detect the onset of the white noise after a gap as short as 2–4 ms in humans, at least after some training [25]. This depends on the ability of the IHC synapse to reliably induce a precisely timed postsynaptic action potential at the onset of the white noise after the silent interval, which will require readily releasable synaptic vesicles. Although this has not been systematically tested in patients with mild forms of DFNB9, we expect that silent gaps will need to be substantially longer to be detected by the probands [26]. On average, animal models with the p.Ile515Thr mutation required 17 ms silence (interpolated value, [13]) to detect the gap, whereas normal hearing mice can perceive gaps as short as 1–2 ms [27].

The combination OAE recordings and ABR or pure tone audiometry are, in principle, sufficient to diagnose hearing impairment due to *OTOF* mutations. Other tests do not provide additional information as, for example, even at high sound pressure levels (SPLs), no auditory reflexes can be elicited, which confirms the absence of auditory evoked signal transmission indicated by absent signals in ABRs. In only a few cases, transtympanic electrocochleography (ECochG), with a recording electrode placed at the promontory wall, will be of use for diagnoses. However, since it requires local anesthesia of the tympanic membrane and is rather invasive, it is questionable if this justifies the limited additional information. ECochG is employed to record the summating potential (SP), cochlear microphonics and compound action potentials (CAPs). Cochlear microphonics originate from functional OHCs such that this recording would be redundant to OAE tests, although amplitudes of cochlear microphonics can be highly variable [23,28].

Recording the SP in response to click stimuli may add novel information in particular cases when OAEs are absent, but might be hard to interpret. Since the SP derives from the depolarization of inner and outer hair cells [29], the depolarization of the IHCs may still result in a small but measurable SP even if OHCs are degenerated. This can help to distinguish from forms of hearing loss involving the stereocilia and/or the mechanotransduction channels, since, in this case, no depolarization of hair cells occurs.

CAPs that record the first action potential in the auditory pathway are absent in some DFNB9 patients, while others exhibit a prolonged CAP with reduced amplitude, at least for single click stimuli [23]. Repetitions of click stimuli with short interstimulus intervals of 2.9 ms abolish CAP responses. Findings from a detailed assessment of pre- and postsynaptic function in the *pachanga* mouse model (p.Asp1772Gly) can likely explain this observation: The IHCs in this animal model display intact synaptic signal transmission for short (<10 ms) interspaced stimuli, given that the interstimulus intervals allow for sufficient recovery [8]. Under repetitive stimulation, as in ABR recordings, the strong defect in vesicle replenishment abolishes reliable signal transmission. In single auditory nerve fiber recordings, the first spike was found to be highly variable in timing and was, on average, delayed, which is likely to reflect the prolonged CAP response. The spike rate in *pachanga* mice reached up to 200 spikes/second (compared to >400 spikes/second in wild-type mice), but only when stimuli were presented once every two seconds (0.5 Hz stimulus frequency) [8]. Increasing the stimulus frequency to 10 Hz strongly diminished neural responses (<10 spikes/second) except for the very first trials. Therefore, we infer that the CAP signals for isolated click stimuli in DFNB9 patients indeed originate from an auditory evoked neural response and are prolonged due to the increased first spike latency. Presenting repetitive click stimuli to these patients—a second stimulus after 15 ms and subsequent ones at 33 Hz—strongly reduced or even abolished these CAP responses, which might be the direct equivalent to the diminished neural spiking found in the mouse models. The reason for this is that the replenishment of synaptic vesicles is strongly slowed down when the amount of otoferlin at the IHC plasma membrane is reduced [8,13]. Thus, long silent intervals are required to regenerate the first auditory synapse to enable another cycle of auditory evoked synaptic transmission.

However, why is the CAP response absent in some DFNB9 patients, or prolonged and with a small amplitude in others? This question arose in a study that analyzed CAP responses in patients with various types of variants in the *OTOF* gene. Two frameshift variants were associated with absent CAPs in two patients [23]. Individuals with biallelic premature stop variants exhibited the largest CAP response in this study, while the amplitude of the CAP was intermediate in patients with one frameshift and one premature stop variant. While frameshift variants cause termination of the amino acid chain in all cases, stop codon read-through can occasionally occur with an efficiency of up to 3–4% (reviewed in [30]). As only 3% of the otoferlin protein is localized at the plasma membrane in *pachanga* mice, it is tempting to speculate that an *OTOF* gene with premature stop codons described in this study may have undergone partial natural stop codon read-through, inducing residual synaptic function of an order of magnitude as in *pachanga* mice.

## 3. Molecular Epidemiology of *OTOF*-Associated Hearing Loss

### 3.1. Summary of Variants Identified in Otoferlin

By virtue of being one of the first deafness genes identified, *OTOF* has been tested in molecular genetic diagnostic settings for over two decades, allowing an estimate of the global burden of *OTOF*-associated hearing loss. There are presently 219 genetic changes that are classified as pathogenic or likely pathogenic according to the literature or clinical database entries (Leiden Open Variation Database v3.0 (LOVD v3), the Deafness Variation Database (DVD), ClinVar, and the Human Gene Mutation Database (HGMD)) (Appendix A). This includes 84 missense, 44 frameshift, 43 nonsense, 36 splice site, 7 in-frame duplications or deletions, 3 copy number variations, as well as 1 stop loss and regulatory variant each (Figure 2 and Figure 3, Appendix A).

### 3.2. Population-Based Diagnostic Rates of Otoferlin

The prevalence of *OTOF*-associated hearing loss varies according to population background. For example, *OTOF* variants account for approximately 5% of genetic diagnoses in the Turkish population [31], and 3.1% of diagnoses in the Pakistani population [32]. A common founder variant (p.Gln829*) was identified in 3% of Spanish cohorts [21,33]. In other populations, *OTOF* has been identified as a cause of hearing impairment in 3.1% of Taiwanese [34], 2.4% (primarily) European-American [35], 2–3% of Pakistani [18,32], 1.9% of French [36] and 1.7% of Japanese [37] patients who were not pre-selected on the basis of auditory neuropathy/synaptopathy. In Iranian patients, a study that included 38 consanguineous patients identified only one family with a homozygous frameshift variant (c.1981dupG, p.Asp661Glyfs*2) and suggested *OTOF* is not a major contributor to hearing loss in the Iranian population [38].

### 3.3. Diagnostic Rates of Otoferlin in Patients with Auditory Neuropathy/Synaptopathy

Auditory synaptopathy with prelingual onset has been identified in patients with genetic aberrations in a small subset of genes (*PJVK*, *OPA1*, and *DIAPH3* (AUNA1 locus)), and a limited number of suspected cases in a few other genes such as *GJB2* [39,40,41,42,43,44], although the *GJB2* cases are controversially discussed [45]. The unique phenotypic presentation of DFNB9 makes a targeted selection for *OTOF* screening in patients for genetic testing rather successful. As exemplified by a study that included Japanese patients with auditory neuropathy/synaptopathy, biallelic *OTOF* variants were uncovered in 56% of cases that included the identification of a founder variant (p.Arg1939Gln) [46]. The p.Gln829* founder variant was identified in 87% of patients diagnosed with auditory neuropathy/synaptopathy in the Spanish population [21]. Another founder variant (p.Glu1700Gln) in Taiwanese patients with progressive, moderate-to-profound hearing loss was identified that diagnosed 23% of a selected patient cohort of 22 individuals with auditory neuropathy/synaptopathy [47]. A study that screened the *OTOF* gene in 37 Chinese patients with congenital auditory neuropathy/synaptopathy had a diagnostic yield of 41.2% [48]. On the contrary, a study that involved the screening of 73 Chinese Han patients with auditory neuropathy/synaptopathy resolved only 5.5% of patients and uncovered a temperature-sensitive variant, which was lower than anticipated and demonstrates a high diagnostic variability [49].

### 3.4. Missing Variants

The diagnostic yield of patients with audiological hallmarks of DFNB9 suggests multifaceted deficits in general isoform and variant knowledge, as well as possible technical limitations. Beyond the possibility of additional genes harboring causally associated variants that evoke the same clinical features, there are several reasons explaining why patients with auditory synaptopathy due to biallelic variants in *OTOF* remain undiagnosed after molecular genetic screening. Such reasons include possible limitations stemming from methodology (e.g., sequencing coverage gaps), missed copy number variations that either fall below the detection resolution of commonly used microarrays in genetic diagnostics or missed due to uneven high-throughput sequencing coverage, especially in the case of exome sequencing, or deep intronic variants that are not captured in targeted enrichment approaches. Furthermore, variant interpretation bottlenecks that could also be due to incorrect transcript usage in variant annotation, current limitations in knowledge about the pathogenicity of rare variants and lack of opportunity for segregation testing in families that can complicate outcomes for definitive statements about variant pathogenicity. Another hypothesis points to variants occurring in currently unannotated exons.

Sequence analysis is primarily focused on exonic regions and relies on the complete understanding of gene isoform structure (i.e., exon annotation). The cochlea is encased in one of the hardest bones of the body, making it one of the least accessible tissues for transcriptome studies. However, many microarray and RNA-seq-based studies using the human and rodent whole cochlea have ensued since the early 2000s [50,51]. Though challenging, single-cell isolation of the inner ear and long read single-cell RNA-seq have recently been performed in mice at several developmental time points [52] to reveal cell-type defining genes and pathways. Long-read sequencing and isoform analysis has identified unappreciated splicing heterogeneity and expression of cell-specific isoforms with unannotated exons [52]. A recent study marked a crucial gap in this understanding in many well-studied genes, such as *Otof*, by mapping a novel non-coding exon 6b and suggesting an in-frame exon 10b (Figure 4). Extending this finding by annotating novel *OTOF* exons in humans could yield significant implications for undiagnosed patients who would otherwise fit the characteristic DFNB9 phenotypic spectrum.

## 4. Genotype-Phenotype Correlations in DFNB9 Patients

The uniformity of available clinical and genetic information about the current set of identified variants is highly variable. For example, reported variant zygosity (i.e., homozygous versus compound heterozygous) and the extent of audiological characterization and recorded onset in patients are highly heterogeneous. Most variants lack recorded audiological information. Generally, biallelic *OTOF* variants cause congenital or early onset (*n* = 114) hearing impairment. Few variants have been identified with progressive hearing loss (*n* = 3). Seven variants have been linked to temperature-sensitive hearing loss, five of which are located within C_2_ domains. While premature stop and frameshift variants typically cause profound prelingual deafness, non-truncating variants can cause a highly variable phenotype. Depending on the localization and the physico-chemical properties of the substituted (or deleted) amino acid residues, variants can severely affect protein stability and contribute to protein degradation. In some cases, the deterioration of protein folding is less severe, leaving some endogenous otoferlin at the plasma membrane that may vary with age and body temperature. This typically results in mildly to moderately elevated thresholds in pure tone audiograms but severely impaired speech comprehension. Notably, patients with point mutations and residual otoferlin expression report perceiving a fading out of a tone burst presented with constant intensity [23,53]. These characteristics of hearing impairment seem to be true for both types of moderate auditory synaptopathy that include the temperature-sensitive and progressive variants.

### 4.1. Temperature-Sensitive Auditory Synaptopathy

Temperature-sensitive auditory synaptopathy has been reported by parents observing profound deafness as soon as their children are febrile. Mirroring daytime changes in body temperature, fluctuations in speech comprehension have been described, which was least affected in the early morning and progressed throughout the day to a point where vocal communication is hardly possible [24]. Even a slight increase in body temperature from 36.5 to 36.8 °C in the course of a day seems to attenuate auditory perception. However, in addition to increased body temperature, acoustic exposure is likely higher during the daytime than at night, which may aggravate hearing impairment due to defective synaptic regeneration. Potentially, both body temperature and increased sound stimulation compound to impair speech comprehension during daytime.

When measured in febrile conditions, both an elevation in pure tone thresholds and zero speech comprehension can confirm the diagnosis of temperature-sensitive auditory synaptopathy. The first variant discovered causing temperature-sensitive auditory synaptopathy, p.Ile515Thr [54], has been extensively studied in a mouse model [13]. At normal body temperature, the hearing phenotype of these mice mirrored what is observed in the affected human siblings. When afebrile, they have tremendous difficulties understanding speech in background noise despite having almost normal pure tone audiograms. In the mouse models, ABR thresholds were mildly elevated, 10 dB SPL for click stimuli and ~20 dB SPL across tone burst frequencies at the age of 3–4 weeks. When the same mice were tested again at the age of 8 and 25 weeks, ABR thresholds increased to an average of 80 dB SPL for tone bursts and to 50 (8 weeks) to ~75 dB SPL (25 weeks) for click stimuli. In parallel, ABR wave amplitudes were strongly diminished. In contrast, behavioral tests and auditory nerve fiber recordings revealed only mild threshold shifts in these mice at that age. This correlates well in patients with impaired ABR despite only mild threshold elevations in psychophysical tests. In mice, single auditory nerve fiber recordings were employed to assess the effect of the presynaptic impairment on action potential generation. In agreement with the presynaptic deficiency in replenishing vesicles, the spike rate in the auditory nerve decreased with longer acoustic stimuli or with upscaling of the stimulus frequency, representing a correlate of the auditory fatigue observed in humans. Moreover, the timing of the first spike after sound onset was of greater variability compared to normal hearing controls. In addition, the phase locking to amplitude modulated tones was strongly impaired. If this timely precision of spiking is lower, click sounds or consonants will become blurred. With respect to human hearing, these deficits most likely explain difficulties in speech comprehension and the abolishment of the latter in background noise. At elevated temperature, patch clamp recordings revealed a decrease in exocytosis when cells were heated from near-physiological (35–37 °C) to elevated temperatures (38.5–40 °C). This was especially obvious for wild-type IHCs, indicating that even the wild-type otoferlin protein is very sensitive to elevated temperature and may unfold rather quickly [13]. However, the wild-type protein seemed to be capable of re-folding, as it gained back initial exocytosis when temperature was lowered to <29 °C. In contrast, the IHCs in the p.Ile515Thr model showed impaired recovery, suggesting that the destabilization of the substitution in the C_2_C domain reduces the likelihood of proper refolding. This interpretation would be in concordance with the observation that patients regain hearing a few hours after temperature-induced deafening, which would be consistent with the time required for *de novo* synthesis of sufficient quantities of otoferlin.

A similar phenotype to the p.Ile515Thr variant was described in humans with the p.Gly541Ser variant, which also localizes to the C_2_C domain of otoferlin [24,46]. In addition, temperature-sensitive auditory neuropathy has been described for the p.Arg1607Trp variant in the C_2_E domain [24,49] and for an individual with compound heterozygosity for the p.Gly614Glu and p.Arg1080Pro substitutions, the latter of which resides in the C_2_D domain [55]. Remarkably, patients with the p.Arg1607Trp substitution in homozygosity or compound heterozygosity reported hearing loss that ameliorated with increasing age [24]. An in-frame deletion, p.Glu1804del in the C_2_F domain, was also attributed to temperature-dependent hearing loss [56]. Notably, no other genes apart from *OTOF* have been associated with temperature-sensitive forms of hearing loss and all cases presenting a similar phenotype in the literature have disclosed pathogenic *OTOF* variants, regardless of population background. All described substitutions seem to cause only slight destabilizations of one C_2_ domain. However, since mutations in different otoferlin C_2_ domains cause a similar phenotype, we consider it unlikely that each of the substitutions causes heat-sensitivity of the protein. Rather, as shown in Strenzke et al. [13], even the native otoferlin protein is considerably temperature sensitive at 38.5–40 °C. Potentially, any slight destabilization of this highly flexible protein might decrease the chance of re-folding after heat exposure, such that more protein is degraded at a slightly elevated body temperature, thereby exacerbating the hearing disturbance.

### 4.2. Progressive Hearing Impairment

Progressive forms of *OTOF* hearing impairment have been described for three variants: p.Ile1573Thr, p.Glu1700Gln and p.Ter1998Argext30Ter [46,47,57]. In all cases, the hearing impairment onset was prelingual, but the severity ranged from mild to profound at onset, even for individuals with the same variant.

The homozygous p.Glu1700Gln substitution was identified in several Taiwanese families [47]. In patients from three families, hearing loss was initially mild and became moderate to severe within a few years. In two other families, affected individuals were identified with severe or profound hearing impairment already at the first hearing assessment at the age of two and one years of age, respectively. The reason for hearing loss progression and variability of the onset severity is currently unknown since the linker region between the C_2_E and the C_2_F domains, in which this substitution lies, has not been studied so far.

A homozygous p.Ile1573Thr substitution was identified in a child with parental consanguinity, in whom hearing deterioration correlated with age [57]. This substitution in the 6th β-strand of the C_2_E domain likely reduces the stability of protein folding. The four children of this family were found to have mild (9 years of age), moderate (11 y, 13 y) or severe hearing loss (17 y). All children displayed OAEs. Absence of ABR waves in the 9-year-old child (the only one tested) is in concordance with the severe abnormality described for all DFNB9 patients, even those with only mildly elevated hearing thresholds. A follow-up of the progression of hearing impairment of this family has not been performed so far.

A stop loss variant p.Ter1998Argext30Ter associated with progressive hearing impairment was found in compound heterozygosity with the p.Arg1939Gln substitution [46]. Since the latter causes profound hearing loss in homozygosity, the early onset, moderate hearing loss, with a steeply sloping audiogram in one ear and a gently sloping audiogram in the other, is presumably due to the elongation of the C-terminus. The C-terminal TM domain (amino acids 1964–1984), as well as the 13 amino acids more downstream, are highly sensitive to substitutions (see Section 5.)

In summary, progression of hearing impairment was typically observed over the course of a few months or years, such that affected individuals reached profound deafness in the second decade of life. Presumably due to the residual otoferlin function, OAEs remained preserved in these intermediate forms of hearing impairment; thus, affected individuals may be candidates for gene therapies even in adulthood.

## 5. Localization and Presumed Effects of Single Amino Acid Substitutions in Otoferlin

Many of the non-truncating variants affect the C_2_, FerA and TM domains, meriting a broader discussion about domain functions and deteriorating effects due to substitutions. C_2_ domains are globular domains comprised of eight antiparallel β-strands, many of which bind phospholipids and Ca^2+^. Since the Ca^2+^-binding site is localized in the structure to five specific aspartate residues in two top loops, the Ca^2+^-binding ability can be reasonably predicted from the sequence. With respect to the C_2_A domain, the structure of the rat protein, which is 91% identical and 96% similar to the human otoferlin C_2_A domain, has been solved with X-ray crystallography [58]. Since only one aspartate is present in the two top loops at respective positions, the C_2_A domain was predicted not to bind Ca^2+^, which was confirmed experimentally [58,59,60].

Different from substitutions in the other otoferlin C_2_ domains, neither amino acid replacements in the β-strands nor in the loops of the C_2_A domain cause malfunction of the protein. The reasons for this might be first, that the C_2_A domain folds much more stably than the other C_2_ domains. Crystallization from heterologous expression has been successful for the C_2_A domain. However, despite laborious efforts from several research groups, this has not been the case for other C_2_ domains so far, presumably because higher protein dynamics prevent the forming of crystals. Notably, the same seems to be true for the myoferlin and dysferlin C_2_ domains, of which only the structures of the C_2_A domains could be resolved to date [61]. Whether the high flexibility of the ferlin C_2_B-C_2_F domains is a biological defect or a feature relevant for proper function remains to be determined. The second reason why substitutions may be tolerated well in the C_2_A domain is that this domain does not bind Ca^2+^ or phospholipids [58,59,60] and thus might be not directly involved in the process of Ca^2+^-triggered vesicle fusion or Ca^2+^-dependent vesicle replenishment.

Prediction of Ca^2+^-binding sites or the location of pathogenic substitutions in the other C_2_ domains is rather challenging due to the low sequence similarity of the otoferlin C_2_B to C_2_F domains to C_2_ domains with known structure. Automatic domain annotation algorithms such as SMART (EMBL, Heidleberg, Germany) [62] do not predict the extent of these domains reliably. We therefore employed Phyre2 to predict the structures of these domains, which is based on homology modelling and makes use of all structures in the protein data bank (PDB) database [63] (Figure 5). Within the predicted structures, we mapped the substitutions causing profound deafness (shaded in orange), and substitutions causing milder forms of hearing loss (shaded yellow/red; Figure 5). With the exception of the 7th and 8th β-strand of the C_2_E domain and the 7th β-strand of the C_2_F domain, all β-strands could be localized in the predictions. The first top loop connecting β-strands 1 and 2 and the third top loop between β-strand 5 and 6 comprise five aspartate residues (blue fonts) that coordinate one to three Ca^2+^ ions. Consistent with experimental data revealing that the C_2_A domain does not bind Ca^2+^, its first top loop misses the motif containing the first aspartate, and, in the third top loop, the three aspartate sites are replaced by neutral or positively charged amino acids. Similarly, the C_2_B domain comprises only one aspartate residue in the top loops, indicating that this C_2_ domain cannot bind Ca^2+^. Consistent with this prediction, one lab found no Ca^2+^ binding for the C_2_B domain using microscale thermophoresis assays [59]; however, other labs did find indications of Ca^2+^ binding with other tests (e.g., [60]).

The same is true for the C_2_C domains, for which some labs found Ca^2+^ and phospholipid binding, while one other lab found that this domain binds Ca^2+^ only after including a phosphomimetic mutation in the first top loop (replacing the blue shaded threonine in the first loop by a glutamate, since this threonine is a site for activity-dependent CaMKIIδ phosphorylation [59]). The actual structure prediction reveals a very long top loop 1, even slightly longer than the one on the PKCα C_2_ domain, comprising a predicted (otoferlin C_2_C) and a confirmed (PKCα) α-helical region. The PKCα C_2_ domain does bind phospholipids, but not Ca^2+^. For the otoferlin C_2_C domain, one aspartate resides in the first top loop and two aspartates to a short top loop three, allowing no clear prediction whether this domain can bind Ca^2+^. We presume that the Ca^2+^ binding of this domain likely depends on posttranslational modifications such as phosphorylation and the direct domain environment, which could be phospholipid membranes, interacting proteins, or both. In a mouse model in which two aspartate residues in the C_2_C domain were replaced by alanine residues, exocytosis appeared to be slightly slowed [12]. This finding would be consistent with such a context-dependent Ca^2+^-binding ability of this domain, but also with interfered phospholipid binding due to altered domain folding.

The C_2_D and C_2_E domains exhibit the five canonical aspartate residues at respective positions in the top loops, and experimental data confirm that both domains bind Ca^2+^ and phospholipids [60]. Nevertheless, for the C_2_E domain, the precise localization of the last two β-strands could not be predicted with the algorithm and the current dataset of structures. Since a growing number of structures is solved and deposited in databases, future structure predictions might result in a reasonable model of the position of the two β-strands. Pathogenic single amino acid substitutions in these domains typically lie within β-strands (Figure 5), likely destabilizing the structure of the domains. The effect is nicely demonstrated in an ENU-mutagenesis induced mouse line, *deaf5Jcs*, with a p.Ile318Asn substitution in the 5th β-strand of the C_2_B domain (indicated with green shading in Figure 5). Immunohistochemical analyses of mouse IHCs revealed almost complete absence of the protein, despite mRNA transcripts were present [64], most likely because misfolding of one domain leads to proteasomal degradation of the mutated protein.

Structure predictions have hinted to a seventh C_2_ domain, termed C_2_de, between the C_2_D and the C_2_E domains that spans amino acids 1143–1220 according to the *Pfam* algorithm. Due to the low sequence similarity and the rather short length of this predicted domain, it is currently unclear if this region folds as a C_2_ domain at all. One frameshift and three splice site mutations have been found in this potential domain, but so far no non-truncating pathogenic variant (Appendix A).

The C_2_F domain seems to be the most unconventional and most susceptible to alterations ultimately leading to protein malfunction. There are presently 18 reported pathogenic/likely pathogenic variants mapping to this domain.

The first top loop comprises eight negatively charged amino acid residues that could potentially contribute to Ca^2+^ co-ordination. Three aspartate residues reside in the canonical positions of the third top loop. Substitution of one of the aspartates in the first top loop (p.Asp1750His) and two in the third top loop (p.Asp1834Asn and p.Asp1842Asn) are each pathogenic. Accordingly, Ca^2+^ co-ordination seems plausible and has experimentally been confirmed [59,60], despite the fact that the precise folding of the first loop is unable to be predicted. Since these Asp>Asn substitutions in the third loop do not change the hydrophobicity, we presume that the Ca^2+^ affinity is strongly reduced by these substitutions, indicating that Ca^2+^ binding to the C_2_F domain is essential for proper function.

Different from the other otoferlin C_2_ domains, where only two amino acids form the bottom loop between β-strands 2 and 3, the loop in the C_2_F domain is predicted to be longer and consists of hydrophobic tryptophane residues flanked by positively charged side chains. This loop comprises the p.Asp1772Gly mutation found in *pachanga* mice which strongly reduced plasma membrane association of otoferlin [8,13,14]. This indicates that this loop structure, also found in other ferlin C_2_F domains [65], seems to be crucial for a partial insertion into phospholipid membranes. Despite homology modelling with alignments with different structures leaving some uncertainty about the beginning of the subsequent β-strand, the consensus prediction of this β-strand includes the position of three consecutive amino acids whose substitutions cause profound hearing loss (p.Asp1777Gly, p.Val1778Ile, p.Val1778Phe, p.His1779Tyr). Moreover, these three amino acids reside just before a CaMKIIδ phosphorylation site, S1782, and might interfere with the binding of the CaMKIIδ and thus phosphorylation [59]. The subsequent bottom loop comprises two glutamate residues. The deletion of one (p.Glu1804del) causes temperature-sensitive auditory neuropathy, presumably because the shortening of this loop destabilizes the domain. Thus, it seems as if the C_2_F domain is especially sensitive to point mutations and that Ca^2+^ and phospholipid binding are crucial for proper protein function.

The function of the FerA domain for synaptic transmission is presently less clear. Small-angle X-ray scattering and in vitro experiments indicated that this domain is comprised of four α-helices, which are connected by a dynamic linker region [66]. The FerA domain binds to phospholipid membranes that is enhanced by Ca^2+^. Four non-truncating substitutions have been identified in the FerA domain so far, but three of those could not be linked to a phenotype (heterozygous, or no reference). The fourth homozygous substitution found in a Taiwanese family alters the second helix (p.Leu760Pro). This suggests that misfolding of the FerA domain is not tolerated, indicating a role for the FerA domain for synaptic transmission that requires further studies.

Substitutions of amino acids in the TM domain cause variable severities of hearing loss. The homozygous p.Pro1987Arg substitution causes early onset, severe to profound hearing impairment, in this specific case, with a bowl-shaped audiogram [19] (Appendix A). A three base pair deletion (p.Leu1967_Lys1968delinsGln) at the TM domain caused early onset, mild hearing loss, similar to the p.Ter1998Argext30Ter [46]. This is likely due to the mechanism by which this tail-anchored structure is inserted into the membrane. Once translation of the amino acid chain has been completed, the C-terminal amino acids bind to the chaperone TRC40 and this complex is targeted to the TRC40 receptors WRB and CAML, which insert the C-terminus into the phospholipid membrane [67,68]. These mutations likely interfere with this tail insertion mechanism, thereby reducing the amount of otoferlin at the plasma membrane, leading to a moderate hearing impairment. In contrast, variants truncating the amino acid chain behind the C_2_F domain and before the TM domain, such as p.Gln1883Ter or the c.5833delA deletion (p.Ile1945Serfs*4), cause profound deafness, indicating that the TM domain is essential for protein function [48].

## 6. Current and Future Therapies for DFNB9

The established therapy for individuals with severe to profound hearing loss due to otoferlin deficiency is currently cochlear implantation. Since this prosthetic bridges the first auditory synapse, which is the only part of the auditory pathway involved in DFNB9, patients benefit well from these devices and gain good or even excellent speech understanding. However, the most sensitive period for developing the capability to understand spoken language is within the first two years of life. It is, therefore, critical to implant patients as early as possible. This requires an early diagnosis, but most children with severe to profound hearing loss due to variants in *OTOF* pass newborn hearing testing because, in most countries, the assessment of OAEs is the method of choice for screening. These children are typically diagnosed at a later stage after parents report a severe delay in speech development. Knowing that deafness, either due to biallelic variants in *OTOF* or auditory neuropathies/synaptopathies of other etiologies, cannot be reliably diagnosed with OAE screenings, but could be aided with currently available therapies, requires switching newborn hearing screenings to routine ABR testing. This is not only of importance with respect to cochlear implantation, which will yield much better outcomes if implanted earlier, but also with respect to a gene therapy, which is currently under development.

The currently favored gene therapeutic approach involves replacing the defective gene by transducing IHCs with correct cDNA by means of recombinant adeno-associated viruses (AAVs, reviewed in [69]). These viral vectors have a series of beneficial features: they are non-pathogenic, they evoke the least inflammatory response, they do not integrate into the genome under most conditions, and the choice of their surface protein allows the vector to target different cell types. The main disadvantage is that they can transport only up to 4.9 kb of foreign DNA, which needs to be subcloned between two AAV gene sequences called inverted terminal repeats (ITRs) of 145 bp each. The 6 kb cDNA length encoding otoferlin has successfully been transduced into IHCs by dual-AAV-approaches, where the cDNA is split to two AAV genomes [70,71]. The latter form head-to-tail multimers in the nuclei of target cells, thereby assembling the split cDNA. By means of splice donor and splice acceptor sites, the ITRs are excised, and the otoferlin mRNA transcribed from dual-AAVs has been demonstrated as being correct [70]. Studies in *Otof*^−/−^ mice have revealed that such dual-AAV strategies successfully and persistently restored hearing [70,71]. At least three companies have prepared for clinical trials with dual-AAV approaches, the first intending to start in 2021 (Akouos, Boston, MA, USA; Decibel Therapeutics, Boston, MA, USA; Sensorion, Montpellier, France). This causal therapy is expected to result in more natural hearing compared to cochlear implants, overcoming limitations such as poor perception of vocal emotions, poor frequency discrimination, or poor speech comprehension in background noise, just to name a few. DFNB9 is predestined for a gene therapy, as all cells develop normally and are in place at birth. However, OHCs degenerate within a few years after birth, as discussed above. For cochlear implantation, the loss of OHCs is not of relevance; however, gene therapy will only yield good outcomes with intact OHC-driven cochlear amplification. Hence, a gene therapy for profoundly deaf DFNB9 patients will need to be applied ideally within the first year of life, both to have OHCs still present and to be within the sensitive time window for language acquisition.

Especially with respect to the envisioned gene therapy, the use of hearing aids for rehabilitation of severe to profound hearing impairment should be critically evaluated. While power hearing aids have successfully induced behavioral responses in DFNB9 children with severe to profound deafness [23], we have to assume that the use of hearing aids is unlikely to assure proper speech comprehension. Studies from animal models indicate that challenging the synapse with a higher rate of acoustic stimuli or more intense stimuli will cause a faster depletion of synaptic transmission. This reduces the capability of the synapse to encode modulations of the input. Moreover, it presumably lowers the timely precision of spiking in the auditory nerve, and thus might blur auditory cues required for speech comprehension.

In addition to being questionable for language acquisition, the use of power hearing aids might accelerate the loss of OHCs, as proposed from observations in retrospective studies [22,72,73]. Whether and potentially why OHCs are more susceptible to noise trauma in DFNB9 patients compared to normal hearing individuals still awaits experimental proof and basic research in animal models. Is the expression of otoferlin in immature OHCs or unknown genetic modifiers related to the loss of DPOAEs (as proposed by [22]) or rather the lack of OHC suppression by efferent inhibition? In intact cochleae, OHCs mechanically amplify the motion of the basilar membrane, which increases the sensitivity of gentle sounds by several orders of magnitude, i.e., they lower hearing thresholds by 50–60 dB SPL. For high SPLs, inhibitory innervation from efferent fibers originating from the medial olivocochlear (MOC) system hyperpolarizes OHCs and thereby suppresses this mechanical amplification. Activation of the MOC efferents occurs through activity in the auditory pathway, which is strongly reduced or missing in absence of otoferlin. Thus, even during exposure to intense sounds, we hypothesize that OHCs do not perceive any inhibitory neurotransmission in DFNB9 patients, as is the case for normal hearing individuals. Chronically high levels of OHC activation, such as, for example, in noise trauma experiments, have been associated with cell death, potentially involving oxidative stress. Thus, power hearing aids should be prescribed and used with caution, potentially only for specialized auditory trainings and not for full day usage.

## 7. Outlook and Conclusions

A timely clinical and molecular genetic diagnosis of *OTOF* hearing impairment should be made as early as possible. Ideally, a clinical diagnosis should occur within the first few days of life if already apparent at birth, with rapid molecular genetic diagnostic results thereafter. Therefore, changes in newborn hearing screening protocols from OAEs to ABRs, or a combination of the two will support an early diagnosis. This will become increasingly important as promising gene therapies emerge. The structure of otoferlin, particularly of the C_2_E and C_2_F domains, is provisionally incomplete based on structural modelling. The possibility of an incomplete overall structure is supported by the identification of novel exons in mouse IHC transcriptome data. Therefore, we recommend genetic re-testing of undiagnosed individuals, especially those with auditory neuropathy/synaptopathy to profit from advances in basic knowledge of isoform structure, as well as improvements in sequencing technologies, bioinformatics approaches, and variant interpretation. Diagnostic laboratories critically rely on annotation of variants to the correct transcript in databases such as LOVD, DVD, ClinVar, and HGMD. In many instances in current versions of these databases, the transcript that is used is incorrect. Therefore, careful attention must be exercised by medical geneticists to report variants with the correct transcript until this can be revised.

## Figures and Tables

**Figure 1 genes-11-01411-f001:**
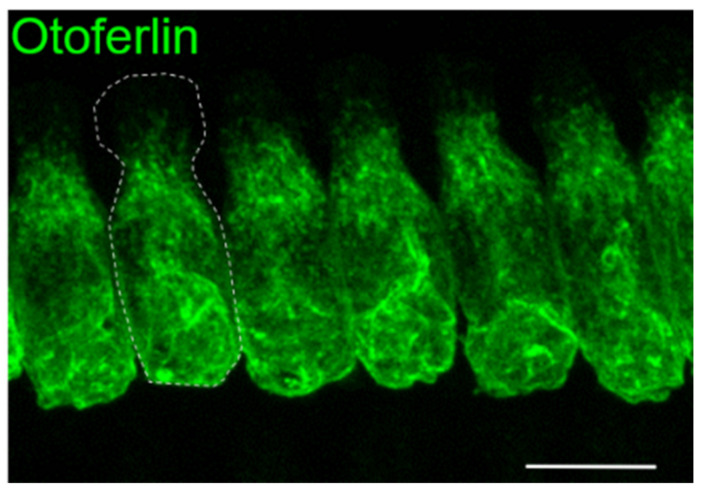
Expression of otoferlin in a row of inner hair cells (IHCs). Maximum projection of optical confocal sections, scale bar: 10 µm (modified from [13]). The dotted white line marks the cell boundary.

**Figure 2 genes-11-01411-f002:**
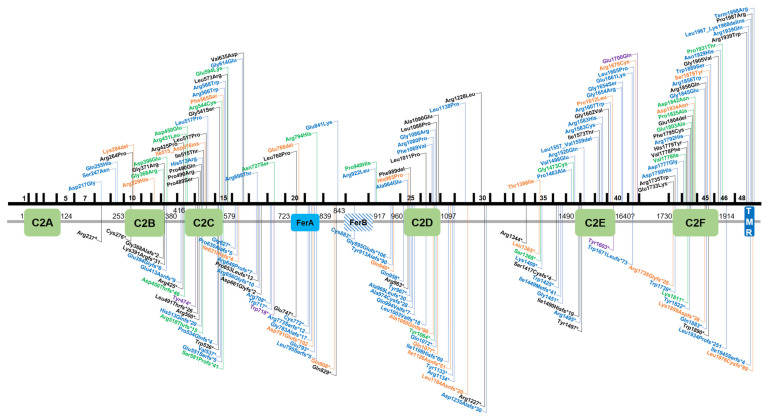
Overview of *OTOF* variants that are classified as pathogenic/likely pathogenic in the databases ClinVar, Leiden Open Variation Database v3.0 (LOVD v3), the Deafness Variation Database (DVD) or Human Gene Mutation Database (HGMD). Variants in the upper part of the figure are non-truncating, variants below are truncating. Black text indicates homozygous variants, blue and green text represents compound heterozygous and heterozygous variants, respectively. Orange text show variants that are reported in databases without a publication reference with undetermined zygosity. Purple text indicates two different variants on the cDNA level that cause the same protein-level change. Variants are annotated according to NM_001287489.1, encoding NP_001274418.1, or isoform e.

**Figure 3 genes-11-01411-f003:**
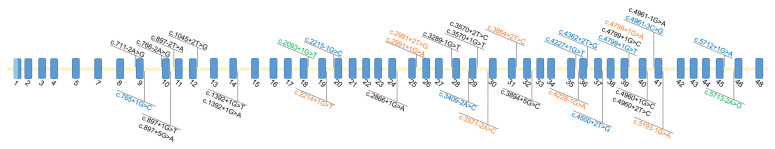
Overview of *OTOF* splice variants according to ClinVar, Leiden Open Variation Database v3.0 (LOVD v3), the Deafness Variation Database (DVD) or the Human Gene Mutation Database (HGMD). Black text indicates homozygous variants, blue and green text represents compound heterozygous and heterozygous variants, respectively. Orange text shows variants that are reported in databases without a publication reference with undetermined zygosity. Variants are annotated according to NM_001287489.1, encoding NP_001274418.1, or isoform e.

**Figure 4 genes-11-01411-f004:**
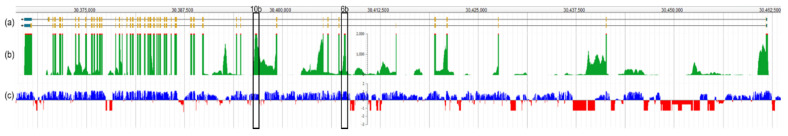
Overview of the *Otof* transcript structure based on single-cell RNA-seq data from mouse IHCs. (**a**) Transcript structure of *Otof*, with exon 1 on the right side of the figure. Exons are depicted as gold bars. Ensembl transcripts *Otof*-201 (upper isoform, encoding 1997 amino acids) and *Otof*-202 (lower isoform, encoding 1992 amino acids) are shown. (**b**) The read depth of each exon is shown in green, with the highest covered regions showing regions in red. Note that maximum read peaks (red) correspond with exons shown in (**a**) if they are expressed in the IHC. (**c**) Mammalian conservation track for *Otof*. Conserved sequences are shown in blue and those not conserved are shown in red. The predicted novel exons 6b and 10b are marked with black boxes. This figure was generated by querying morlscrnaseq.org using the “Transcript Structure Browser” tool [52].

**Figure 5 genes-11-01411-f005:**
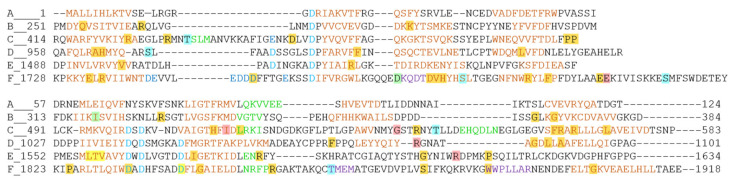
Alignment of otoferlin C_2_ domains A to F with β-strands in brown fonts and α-helices in green fonts. The structure of the C_2_A domain was resolved by X-ray crystallography ([58], PDB accession code 3L9B, www.rcsb.org). The structures of the other C_2_ domains were modelled by means of Phyre2 [63]. The two last β-strands of the C_2_E domain and the 7th β-strand of the C_2_F domain could not be reliably predicted due to low sequence homology. In case the modelling of the β-strands is rather uncertain, amino acids are depicted in purple font. The aspartate residues that coordinate Ca^2+^ are depicted in blue fonts, in case several aspartates or glutamates could potentially play a role for Ca^2+^ co-ordination in dark blue fonts. Pathogenic variants leading to profound deafness are shaded in orange. Those causing moderate hearing loss are shaded in yellow, and if the hearing loss appears to be temperature sensitive, in red. Shading in green indicates mutations in deaf mouse models, *deaf5Jcs* in the C_2_B domain and *pachanga* in the C_2_F domain [14,64]. Threonine or serine residues that were found to be phosphorylated by CaMKIIδ are shaded in blue [59]. Few pathogenic variants affect the CaMKII consensus phosphorylation site, which is RXXS or RXXT.

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
