# Peer review of "The Many Faces of DFNB9: Relating OTOF Variants to Hearing Impairment"

_genes, 2020, doi:10.3390/genes11121411_

Round 1

Reviewer 1 Report

Dear Authors,

the present manuscript thoroughly and critically examines clinical aspects associated with phenotypes variants of OTOF patients, often left undiagnosed, with consequent relevant impact on the diagnostic paths for neonatal hearing screening, a topic much debated today.

I found the work complete, exhaustive and multidisciplinary.

The only remark is to better separate the genetic and clinical parts of the manuscript, in order to facilitate a multi-disciplinary readership (pediatricians, audiologists and clinicians).

Author Response

Reviewer 1

The present manuscript thoroughly and critically examines clinical aspects associated with phenotypes variants of OTOF patients, often left undiagnosed, with consequent relevant impact on the diagnostic paths for neonatal hearing screening, a topic much debated today.

I found the work complete, exhaustive and multidisciplinary.

The only remark is to better separate the genetic and clinical parts of the manuscript, in order to facilitate a multi-disciplinary readership (pediatricians, audiologists and clinicians).

Response: We thank the reviewer for this suggestion. We re-arranged the structure of the review to improve readability. We have divided section three into smaller sub-sections and included the former section 6 (missing variants) into section 3 as section 3.4 now. We feel that the genetic and clinical parts are now better separated.

Reviewer 2 Report

OTOF encoded otoferlin and is an important gene related to auditory neuropathy spectrum disorder. The prevalence of OTOF-related hearing impairment is high in many populations. Both clinical and genetic diagnoses are challenging owing to the diverse phenotypes, genotypes, and molecular structures of otoferlin. Vona and colleagues described the different aspects of DFNB9, including the audiological characteristics, the pathogenic/likely pathogenic variants, and the genotype-phenotype correlations. They also discussed the varieties of the phenotypes and genotypes from the view of structural diversity and molecular function of otoferlin, as confirmed by the mouse models.

In general, this is a well-organized and thorough review detailing the diverse auditory phenotypes of DFNB9 on different audiological tests, genetic diversity of OTOF, and molecular features of otoferlin. The comprehensive review of different aspects of DFNB9, from phenotypic performance, genetic profiles to molecular mechanisms, can help the readers attain an overall concept of DFNB9.

There are several typos that should be revised:

  1. Page 6, lines 243-244, in-frame deletion is duplicated.
  2. Page 9, line 277, the variant should be p.Arg1939Gln.

Author Response

Reviewer 2

OTOF encoded otoferlin and is an important gene related to auditory neuropathy spectrum disorder. The prevalence of OTOF-related hearing impairment is high in many populations. Both clinical and genetic diagnoses are challenging owing to the diverse phenotypes, genotypes, and molecular structures of otoferlin. Vona and colleagues described the different aspects of DFNB9, including the audiological characteristics, the pathogenic/likely pathogenic variants, and the genotype-phenotype correlations. They also discussed the varieties of the phenotypes and genotypes from the view of structural diversity and molecular function of otoferlin, as confirmed by the mouse models. 

In general, this is a well-organized and thorough review detailing the diverse auditory phenotypes of DFNB9 on different audiological tests, genetic diversity of OTOF, and molecular features of otoferlin. The comprehensive review of different aspects of DFNB9, from phenotypic performance, genetic profiles to molecular mechanisms, can help the readers attain an overall concept of DFNB9.

There are several typos that should be revised:

  1. Page 6, lines 243-244, in-frame deletion is duplicated.
  2. Page 9, line 277, the variant should be p.Arg1939Gln.

Response: We thank the reviewer for the feedback and catching the typos. Both typos have been corrected.